# New circulation of genotype V of Crimean-Congo haemorrhagic fever virus in humans from Spain

**Lia Monsalve Arteaga**[1], **Juan Luis Muñoz Bellido**[2], **Ana Isabel Negredo**[3], **Jorge García Criado**[4], **Maria Carmen Vieira Lista**[1], **Jesús Ángel Sánchez Serrano**[4], **María Belén Vicente Santiago**[1], **Amparo López Bernús**[1,5], **Fernando de Ory Manchón**[6], **María Paz Sánchez Seco**[3], **Nuria Leralta**[6], **Montserrat Alonso Sardón**[1], **Antonio Muro**[1]*, **Moncef Belhassen-García**[1,5]*

1 Infectious and Tropical Diseases Group (e-INTRO), IBSAL-CIETUS (Biomedical Research Institute of Salamanca-Research Center for Tropical Diseases at the University of Salamanca), Faculty of Pharmacy, University of Salamanca, Salamanca, Spain, 2 Servicio de Microbiología y Parasitología, Complejo Asistencial Universitario de Salamanca, Salamanca, Spain, 3 Arbovirus and Imported Viral Diseases Unit, Centro Nacional de Microbiología, Instituto de Salud Carlos III, Red de Investigación Colaborativa en Enfermedades Tropicales, Madrid, Spain, 4 Emergency Department, Complejo Asistencial Universitario de Salamanca, Salamanca, Spain, 5 Internal Medicine and Infectious Disease Department, Complejo Asistencial Universitario de Salamanca, Salamanca, Spain, 6 Centro Nacional de Microbiología, Instituto de Salud Carlos III, Majadahonda, Madrid, España; Ciber en Salud Pública (CIBERESP), Instituto de Salud Carlos III, Spain

* ama@usal.es (AM); belhassen@usal.es (MB-G)

**Data Availability Statement:** All relevant data are within the manuscript.

## Abstract

### Background

Crimean-Congo haemorrhagic fever (CCHF) is a widespread tick-borne viral disease caused by the Crimean-Congo haemorrhagic fever virus (CCHFV). CCHFV has been implicated in severe viral haemorrhagic fever outbreaks. During the summer of 2016, the first two cases with genotype III (Africa 3) were reported in Spain. The first objective of our study was to determine the presence of CCHFV among patients with febrile illness during the spring and summer periods in 2017 and 2018. Finally, we perform a phylogenetic analysis to determine the genotype of the virus.

### Methodology

We prospectively evaluated patients aged 18 years and older who came to the emergency department at the Salamanca's University Hospital (HUS) with fever. Specific IgM and IgG antibodies against CCHFV by ELISA and one immunofluorescence assay against two different proteins (nucleoprotein and glycoprotein C) was done. Moreover, molecular detection by Real Time PCR was performed in all collected samples. A phylogenetic analysis was carried out to genetically characterize CCHFV detected in this study.

### Principal findings

A total of 133 patients were selected. The mean age was 67.63 years and 60.9% were male. One-third of the patients presented an acute undifferentiated febrile illness. Three

**Funding:** "This study was supported by the Institute of Health Carlos III, ISCIII, Spain (www.isciii.es) RICET, RD16/0027/0018 and RD16CIII/0003/0003 CNM/ISCIII-D, with European Union co-financing by FEDER (Fondo Europeo de Desarrollo Regional) 'Una manera de hacer Europa' to LMA. The funders had no role in study design, data collection and analysis, decision to publish, or preparation of the manuscript.

**Competing interests:** The authors have declared that no competing interests exist.

patients had anti-CCHFV IgG antibodies, suggesting a previous infection. One patient had anti-CCHFV IgM antibodies and a confirmatory RT-PCR. Phylogenetic analysis indicated that the virus corresponds to the European genotype V. This patient came to the emergency department at HUS in August 2018 presenting an acute febrile syndrome with thrombopenia and liver impairment.

## Conclusions

We describe a new circulation of European genotype V CCHFV in Spain. Moreover, this study suggests that CCHFV is an identifiable cause of febrile illness of unknown origin in Spain. Thus, CCHF could be suspected in patients with fever, liver damage, and/or haemorrhagic disorders, particularly in people with risk activities who present in the spring or summer.

### Author summary

Crimean Congo hemorrhagic fever (CCHF) is a widespread tick-borne viral disease caused by a *Nairovirus* of the *Bunyaviridae* family. CCHFV has been considered to be one of the eight priority emergent pathogens for the last 3 years by the World Health Organization (WHO), requiring urgent attention in Research, Development and Innovation (R&D&I) because of its epidemic potential in the near future In this study, we describe a new circulation of European genotype V CCHFV in Spain. Moreover, this study suggests that CCHFV is an identifiable cause of febrile illness of unknown origin in Spain. Thus, CCHF could be suspected in patients with fever, liver damage, and/or haemorrhagic disorders, particularly in people with risk activities who present in the spring or summer.

## Introduction

Crimean-Congo haemorrhagic fever (CCHF) is a viral infectious disease caused by the homonymous microorganism, the Crimean-Congo haemorrhagic fever virus (CCHFV), a tick-borne virus of the genus Ortho*nairovirus* and the *Nairoviridae* family. This virus is transmitted to humans by infected tick bites and by direct contact with blood or other bodily fluids or tissues of viraemic humans or livestock [1,2]. CCHFV is a negative-sense, single-stranded RNA virus with a 3-segmented genome designated as large (L), medium (M), and small (S), according to size, with high genetic variability and reassortment events have been described as other mechanism to increase genetic diversity [3].To date, 6/7 genetic genotypes have been identified based on the of S RNA segment sequence homology of CCHFV [4]. Genotypes I, II and III are mainly distributed in Africa, genotype IV in Asia, genotype V in East Europe, and genotype VI in Greece [3]. Genotype VI seems to be less pathogenic than the other ones [5].

In Europe, this virus has caused major outbreaks in the eastern region [6] (principally in Balkan countries, Turkey, and Russia). However, in recent years, its epidemiology has been changing, and climate changes has been associated as one of the factors driving the circulation of CCHFV [7]. CCHF is currently considered endemic in areas of Southwest Europe since six human cases have been identified in western Spain since the summer of 2016 with 2 first cases detected being one of them by nosocomial transmission [8,9]. In 2018 a new CCHF human

case occurred in Spain associated with handle wild animals and in 2020, 3 new CCHF cases has been diagnosed all of these associated with tick bite. The CCHFV genotype described from patients in 2016 belonged to African genotype III., This genotype is widely distributed in Africa and recently it has been detected in ticks on migratory bird in Morocco, then one of the possible CCHFV entrance routes in Spain is through migratory birds carrying premature forms of infected ticks from Africa [10]. The CCHFV genotype associated with the human case in 2018 consisted in a reassortant virus in S segment assuming a second introduction of CCHFV in the Iberian peninsula, Currently, CCHFV enzootic cycle has been established in some areas of Spain with infected ticks identified mainly in wild animals [11] and with high seroprevalence rates in domestic and wild animals, which increases the risk of CCHF human cases and it is necessary a greater awareness in the diagnostic suspicion.

From a clinical point of view, CCHF can present with a wide clinical spectrum, from an asymptomatic or oligosymptomatic disease to a life-threatening infectious condition with fever, vomiting and diverse haemorrhagic manifestations that may lead to multi-organ failure and death [12]. Laboratory analyses are frequently altered, with leukopenia, thrombocytopenia and elevated transaminases [13]. A strong clinical suspicion is needed in order to obtain a clear and fast diagnosis, initiate supportive treatment if needed, and activate biosafety measures to prevent nosocomial transmission [10].

The aim of this study was to evaluate whether CCHFV is a cause of acute undifferentiated febrile syndrome, to determine the epidemiologic risk factors and to describe the main clinical and laboratory characteristics of the acute cases identified. In this way, genetically characterization of CCHFV in the samples was done.

## Material & methods

### Ethics statement

This study was approved September 7th, 2017 by the Bioethics Committee of CAUSA with the code Cómite Ético CEIC: PI9109/2017. Verbal consent was obtained from each participant. Also, the procedures described here were carried out in accordance with the ethical standards described in the Revised Declaration of Helsinki in 2013. All clinical and epidemiological data were anonymized.

### Study type and sample collection

Descriptive, cross-sectional study that was carried out during the months of May to October in 2017 and 2018 at the emergency unit of HUS located in western Spain. It covers an area of 12,350 km$^2$ encompassing 362 municipalities with a population of 331,473 individuals. All patients with an age above 18 years who came to HUS for a febrile illness without an aetiological diagnosis were eligible. Patients were evaluated by the emergency department and were included if they presented with fever without an aetiological diagnosis and 133 patients had these characteristics. One-hundred thirty-three serum and plasma samples were taken to determine the presence/absence of CCHFV by serological and molecular techniques. Details were collected from the medical records.

### Immunological techniques

The antibodies for CCHFV were analysed according to methodology previous described [14]. Briefly, we used a commercial immunological kit (Vector Best, Novosibirsk, Russia) according to the manufacturer's instructions. The cut-off was calculated as the mean of the adjusted optical density (OD) of the negative control serum plus 0.2. All samples were tested in triplicate

and the mean values were used for this study. Positives samples were confirmed by an in-house ELISA described in Dowall SD et al. [15], and an immunofluorescence assay (IFA) (Euroimmun, Lübeck, Germany)) using nucleoprotein (NP) and glycoprotein (GPC) antigens at 1:10 dilution was used to confirm the positive results.

## PCR molecular detection techniques

Total RNA was extracted from the plasma samples using the QIAamp viral RNA Mini kit (QIAgen, Germany) according to the recommendations of the supplier. The RNA was eluted in 60 μL of RNase-free water and stored at -80˚C until needed. Furthermore, a real-time poly-merase chain reaction (RT-PCR) described by Atkinson et al. [16] and slightly modified in our laboratory to incorporate an internal control consisting in a heterologous DNA fragment of 135 pb flanked by nucleotide primer pair included into reaction mix, for amplification was used for the detection of the CCHFV genome in all the samples and to confirm CCHFV posi-tive results we performed a RT-nested PCR assay described by Negredo et al 2017 [8]. The RT-nested-PCR assayed was designed in the same gene (S) as the screening method but in a differ-ent genomic region.

## Positive acute or previous cases of CCHFV

Acute cases were defined as the presence of confirmed IgM (by at least two different assays) and/or positive RT-PCR. Previous cases were characterized by the presence of IgG antibodies confirmed by two of the performed tests in the absence of markers of acute infection. Cases with a single IgG positive result, not confirmed by any other assay, were considered as indeterminate.

## DNA sequencing

Two fragments of 527pb and 314 pb from S viral segment were obtained with two RT- semi nested PCR assays applying the method described by Negredo et al [8]. We purified amplified DNA by using an Illustra ExoProStar Kit (GE Healthcare Life Sciences, https://www.gelifesciences.com). Double-stranded DNA was sequenced directly by using the Sanger chain-termination method and the BigDye Terminator v3.1 Cycle Sequencing Kit Protocol and the ABI PRISM 3700 DNA Analyzer (Applied Biosystems, https://www.thermofisher.com). We used the sequencing primers of the nested PCR [8]. Sequences of each fragment were assem-bled and analysed by using the SeqMan Program in the Lasergene Package (https://www.dnastar.com) obtaining a consensus fragment 488 pb in size.

## Phylogenetic analysis

Sequences were aligned (Muscle into MEGA X software) and a phylogenetic tree was con-structed by the Neighbor-Joining (NJ) method based on partial (488 nt) sequence of the CCHFV S segment using MEGA X software. Bootstrap confidence limits were based on 1000 replicate.

## Statistical analysis

All the data were statistically analysed using the SPSS Statistics 23.0. software (*Statistical Pack-age for the Social Sciences*). Proportions were calculated for the qualitative variables and stan-dard deviation (SD) and interquartile range (IQR) was calculated for the mean and median respectively.

**Table 1. Main epidemiological and clinical characteristics of patients.**

| Characteristics | N = 133, n (%) |
|---|---|
| Age mean ± SD, years | 67.6 (18.8) |
| Median age, years | 73 (IQR, 54.5–82) |
| Male gender | 81 (60.9) |
| Urban population | 101 (68.7) |
| **Emergency department initial diagnosis** | |
| Respiratory syndrome | 37 (27.8) |
| Genito-urinary syndrome | **32 (24.0)** |
| Febrile syndrome without focus | 30 (22.5) |
| Fever after tick exposure | 8 (6.0) |
| Neurological syndrome | 6 (4.5) |
| Gastrointestinal syndrome | 6 (4.5) |
| Biliary and hepatic infection | **4 (3.0)** |
| Cutaneous affectation | 4 (3.0) |
| Mononucleosis syndrome | 2 (1.5) |
| ENT infection | 2 (1.5) |
| Fever associated to haemodialysis process | 2 (1.5) |

ENT: Ear, nose, and throat disorder

## Results

One-hundred thirty-three patients were included in this study. The main epidemiological and clinical data are shown in **Table 1**. The mean age (±SD) was 67.63 years (±18.8), and 81 (60.9%) were male. Most of these individuals presented with respiratory and genitourinary symptoms and were diagnosed with nonspecific febrile syndrome in the emergency department from HUS.

Serological and molecular CCHF diagnostic assays showed that seven patients had positive results in some test. One patient (Case 1) had anti-CCHFV IgM antibodies by two serologic assays. This patient also had positive results by PCR methods (**Table 2**).

**Table 2. Serological and Real Time RT-PCR results.**

| Case | ELISA IgG | | IFA (IgG) | | ELISA IgM | | IFA IgM | | Real Time RT-PCR | Final classification |
|---|---|---|---|---|---|---|---|---|---|---|
| | Vector Best | NCM[*] | GP[**] | NP[***] | Vector Best | NCM | GP | NP | | |
| 1 | 0.2 (-) | 2 (+) | Negative | Negative | 5.5 (+) | 0.7 (-) | Negative | Positive | Positive | Confirmed acute infection |
| 2 | 10 (+) | 2.5 (+) | Negative | Negative | 10.2 (+) | 0.8 (-) | Negative | Negative | Negative | Confirmed previous infection (Undetermined acute infection) |
| 3 | 10 (+) | 9 (+) | Negative | Negative | 0.02 | | | | Negative | Confirmed previous infection |
| 4 | 1 (+) | 1.3 (+) | Negative | Negative | 0.01 | | | | Negative | Confirmed previous infection |
| 5 | 3.3 (+) | 0.9 (+/-) | Negative | Negative | 0.01 | | | | Negative | Indeterminate previous infection |
| 6 | 1.7 (+) | 0.6 (-) | Negative | Negative | 0.01 | | | | Negative | Negative |
| 7 | 1.9 (+) | 0.1 (-) | Negative | Negative | 0.03 | | | | Negative | Negative |

[*] NCM: National Centre of Microbiology

[**] GP: Glycoprotein

[***] NP: Nucleoprotein

Case 2 also presented IgM positive results by VectorBest EIA, but not confirmed by the other performed analysis and the result by PCR was negative. However, cases 2 3 and 4 had anti-CCHFV IgG confirmed by 2 or more of the performed assays, indicating previous infection. Finally, 3 patients (case 5, 6 and 7) presented a positive result for IgG in one assay while an undetermined result was obtained in another assay, being classified as indeterminate.

In case 1, the PCR confirmatory assay amplified a fragment of S genomic segment and the sequences showed 98% identity with Russian 4495-ST-2008 strain of European origin. The phylogenetic tree based on this sequence grouped with European genotype V (Fig 1), thus a new genotype was detected in human in Spain. Complete genome sequence was not possible because of low amounts of sample but we obtained with Next Generation Sequencing techniques short fragments of M and L segments that showed 98% identity with Russian 1231-CR/TI-2015 strain and 99% with Kosova Hoti strain respectively both belonged to genotype V. This data confirms that the new CCHFV detected in human is not result from reassortment events, all 3 genomic segments belonged to genotype V indicating that the samples from this case (Salamanca Spain 2018, case 1) belong to European genotype V. It is the first time that this genotype is described in human infected with CCHFV in Western Europe.

The main clinical data of patients with acute and previous CCHF are described in Table 3.

The patient with confirmed acute CCHF was a 53-years-old man, involved in cattle husbandry in Béjar, Salamanca province (coordinates: 40.38641 latitude -5.76341 longitude), a small city with 12,961 inhabitants, near to the Portuguese border. He presented in the emergency department of HUS at the beginning of August 2018 with a history of fever of 5 days, chills, mouth ulcerations (but not any haemorrhagic oral bullae) and acute leg myalgias with no bleeding symptomatology. Laboratory analysis revealed leukopenia, thrombopenia, increase of transaminases with an anicteric cholestasis, and prolongation of activated partial thromboplastin time. Also, a hemophagocytic syndrome was raised in order of the presence of hyperferritinemia (>10,000 ng/mL), hypertriglyceridemia and increase of lactate dehydrogenase (LDH). Also, it is important to asseverate that no nosocomial cases were reported.

## Discussion

In this study a new acute CCHF case has been detected in Spain, being 7 CCHF cases since the causative agent was discovered in this country. CCHFV was found in Western Spain for the first time in 2010, in ticks (*Hyalomma lusitanicum*) feeding on wild animals in the province of Cáceres [17] bordering on the north with the province of Salamanca, where this study was performed. The first human autochthonous case of CCHF was reported in this geographical zone six years ago, and even so, until now the diagnosis of this disease was not suspected [8]. Surveillance studies on West, South and Central areas of Spain, all with *Hyalomma* spp presence, showed endemic CCHFV circulation [11,18] in this European region and thus increasing the risk of illness cases [11].

This study was performed to determine whether CCHFV could be a cause of acute non-specific febrile syndrome in Spanish emergency departments. To our knowledge, this study is the first conducted in Spain with this purpose. Our results showed that seven of the 133 enrolled patients presented at least one diagnostic marker for ancient contact with the CCHFV, and this result could be confirmed by a second serologic method in four of them. In terms of the impact of the disease, we found an elevated incidence and prevalence, with 0.75% (1/133) of acute patients who were IgM-positive and with positive RT-PCR test and an additional 2.2% (3/133) of patients who were IgG positive, suggesting previous infections. According to these results, testing for CCHFV in Spain, should be performed in geographical areas with CCHFV enzootic cycles, Central South western Spain where other tick-borne diseases such as Lyme

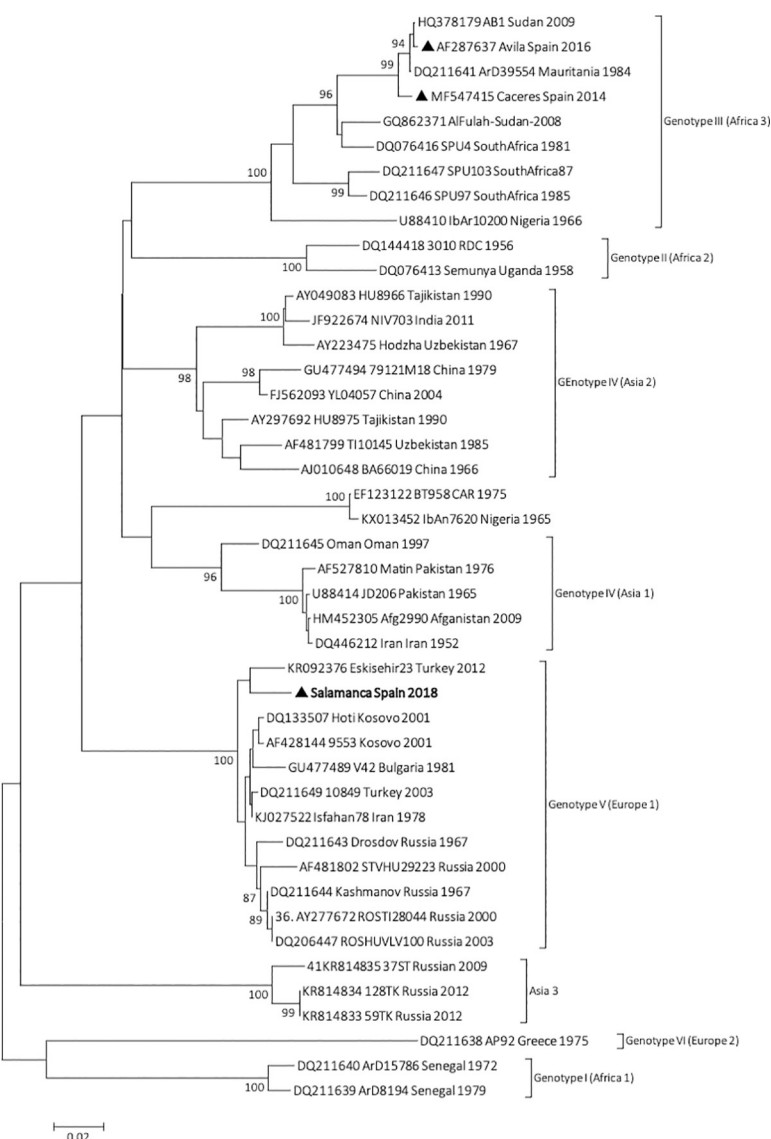

**Fig 1. The phylogenetic tree was constructed by the neighbor-joining (NJ) method and based on partial (488 nt) sequences of the CCHFV S segment.** Numbers in the right indicate bootstrap values for the groups. Triangles indicate Spanish strain and the new CCHFV strain Salamanca 2018 described here is shown in bold and are named by locality sampling site, geographic origin, and sampling year; other sequences are named by GenBank accession number, strain, geographic origin, and sampling year. Scale bar indicates nucleotide substitution per site. Genotypes are indicated in roman numerals, named according to (Carrol 2010) with the equivalent clade nomenclature according to (Chamberlain 2005) indicated in brackets: I-West Africa (Africa 1); II-Central Africa (Africa 2); III-South and West Africa (Africa 3); IV-Middle East/Asia, divided in two groups corresponding to groups Asia 1 y Asia 2; V-Europe/ Turkey (Europe 1); VI-Greece (Europe 2).

disease, tularaemia, babesiosis or anaplasmosis, that have a lower incidence, are suspected [19,20]. The viral nucleic acids amplified in the blood of the patient showed the greatest genetic identity with viruses of the European genotype V, a recently genotype described in ticks in Spain [18] instead of African 3 genotype described previously [8,12]. The new CCHFV genotype V detected in ticks and human in two distant geographical area reveals the expansion of this virus and the possible establishment of a transmission cycle of CCHFV genotype V in our country. It was not possible to obtain complete genome sequence of this viruses probably due

**Table 3. Main clinical and laboratory data of patients with CCHFV.**

| | Acute infection by CCHFV, N = 1 | Previous infection by CCHFV, N = 3 |
|---|---|---|
| Age mean ± SD, years | 53 | 68 ± 28.2 |
| Sex | Male | 2 males / 1 female |
| Urban population | Yes | 2/3 |
| Comorbidity | Non | 2/3 arterial hypertension<br>1/3 chronic renal failure, hypothyroidism, depressive syndrome, dyslipidemia and stroke<br>1/3 essential thrombocytosis |
| First clinical diagnosis | Acute viral hepatitis | 2/3 genitourinary tract infection<br>1/3 acute pancreatitis |
| Range fever duration, days | 5–10 | 3–7 |
| Chills | Yes | 2/3 |
| Abdominal pain | Non | 1/3 |
| Cutaneous signs (suggesting tick bite) | Leg sore | 0 |
| Muscles soreness | Leg myalgias | 1/3 |
| Any bleeding symptomatology | Non | Non |
| Risk factors | Cattle husbandry | Non |
| **Laboratory data** | | |
| **Hemogram, ±SD** | | |
| Haemoglobin, g/dL | 14.1 | 9.6 ± 2.4 |
| White blood cells, x $10^3$/mm$^3$ | 3.1 | 7.0 ± 1.9 |
| Polymorphonuclear leukocytes x $10^3$/mm$^3$ | 8.1 | 5.4 ± 1.5 |
| Lymphocytes x $10^3$/mm$^3$ | 3.6 | 1.1 ± 5.6 |
| Platelets, x $10^3$/mm$^3$ | 41 | 207.6 ± 158.5 |
| **Liver function tests, ±SD** | | |
| C-Reactive Protein (CRP), mg/L | 15.16 | 20.2 ± 15.5 |
| Activated Partial Thromboplastin Time, sec | 43.8 | 34.2 ± 5.1 |
| Aspartate Aminotransferase (AST), U/L | 347 | 151.3 ± 110.2 |
| Alanine Aminotransferase (ALT), U/L | 161 | 74.7 ± 50.1 |

to the low viral load in the sample. The cycle threshold for this sample was >30 and it is described that mild illness cases can have low viral load [21].

Our positive acute case was a middle-aged male, involved in livestock husbandry, without a history of travel to Eastern Europe. Clinical characteristics were a history of fever, chills, myalgias, with thrombopenia and prolongated coagulation times, with no evidence of bleeding or hepatomegaly. These features are similar to those found in previous studies conducted in other endemic countries such as Georgia and Bulgaria [21–22]. The main differences found between our patient with acute CCHFV and those reported in other endemic countries were the lack of any bleeding history or bleeding stigma and the lack of hepatomegaly or splenomegaly at the clinical examination.

According to our data, patients with CCHFV infections could present to the emergency department with a febrile syndrome, with thrombopenia, prolongated coagulation times, elevated transaminases levels, even without any haemorrhagic signs or symptoms. Spanish physicians should have this clinical suspicion when they face a patient with a similar clinical picture with history of contact with animals or ticks in geographical areas with CCHFV enzootic cycles, Central and South western Spain, and mainly especially in the period between May and October when the vectors of the disease, Hyalommas spp have active transmission cycles. All

of this in order to rapidly start the supportive and biosafety measures, to avoid complications linked to the patient morbidity, and the possible nosocomial outbreaks [12,23].

Our hypothesis is that this novel variant of the virus was introduced in the Spanish territory by livestock displacement through (legal or illegal) animal trade from zones of East, Centre Europe to Spain. Although, it is known that African III genotype was introduced in our territory by migratory birds from Western Africa [10]. So far, principal direction of bird migration is north-south, and birds from Eastern countries of Europe fly in autumn through Turkey to reach Africa, however, the possibility of the introduction of the genotype V by migratory birds cannot be completely ruled out. Further studies should be carried out in order to clarify how this novel strain has reach the Spanish territory.

## Conclusion

A CCHFV of genotype V was described in human from Spain in a retrospective new case of CCHF. This study suggests that CCHF is an identifiable cause of non-specific febrile illness in Spain; therefore, it is mandatory to suspect this disease when a patient comes to the emergency department with fever, thrombocytopenia and transaminase elevation, especially in spring and summer, and when patients have an occupational risk, following the protocols already established for this purpose. All of this is intended to initiate supportive treatment and isolation measures as soon as possible, thus reducing the mortality risk and avoiding the risk of a nosocomial outbreak.

## Author Contributions

**Conceptualization:** Lia Monsalve Arteaga, Ana Isabel Negredo, Antonio Muro, Moncef Belhassen-García.

**Data curation:** Lia Monsalve Arteaga, Juan Luis Muñoz Bellido, Maria Carmen Vieira Lista, Montserrat Alonso Sardón, Moncef Belhassen-García.

**Formal analysis:** Lia Monsalve Arteaga, Juan Luis Muñoz Bellido, Montserrat Alonso Sardón, Antonio Muro, Moncef Belhassen-García.

**Funding acquisition:** María Paz Sánchez Seco, Antonio Muro, Moncef Belhassen-García.

**Investigation:** Lia Monsalve Arteaga, Jorge García Criado, Jesús Ángel Sánchez Serrano, Amparo López Bernús, María Paz Sánchez Seco, Nuria Leralta.

**Methodology:** Lia Monsalve Arteaga, Juan Luis Muñoz Bellido, Montserrat Alonso Sardón, Antonio Muro, Moncef Belhassen-García.

**Project administration:** Antonio Muro, Moncef Belhassen-García.

**Resources:** Ana Isabel Negredo, María Belén Vicente Santiago, Fernando de Ory Manchón, María Paz Sánchez Seco, Antonio Muro, Moncef Belhassen-García.

**Software:** Lia Monsalve Arteaga, Maria Carmen Vieira Lista, Montserrat Alonso Sardón.

**Supervision:** Juan Luis Muñoz Bellido, Ana Isabel Negredo, Montserrat Alonso Sardón, Antonio Muro, Moncef Belhassen-García.

**Validation:** Juan Luis Muñoz Bellido, Jorge García Criado, Jesús Ángel Sánchez Serrano, Antonio Muro.

**Visualization:** Antonio Muro, Moncef Belhassen-García.

**Writing – original draft:** Lia Monsalve Arteaga, Antonio Muro, Moncef Belhassen-García.

**Writing – review & editing:** Lia Monsalve Arteaga, Antonio Muro, Moncef Belhassen-García.

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
