## [Decision Letter · Decision Letter 0]

13 Jan 2021

Dear Dr. Belhassen-García,

Thank you very much for submitting your manuscript "New circulation of genotype V of Crimean-Congo haemorrhagic fever virus in humans from Spain." for consideration at PLOS Neglected Tropical Diseases. As with all papers reviewed by the journal, your manuscript was reviewed by members of the editorial board and by several independent reviewers. The reviewers appreciated the attention to an important topic. Based on the reviews, we are likely to accept this manuscript for publication, providing that you modify the manuscript according to the review recommendations. 

Sincerely,

Dennis A. Bente, D.V.M., Ph.D.

Deputy Editor

Dennis Bente

Deputy Editor

Reviewer's Responses to Questions

**Key Review Criteria Required for Acceptance?**

**Methods**

-Are the objectives of the study clearly articulated with a clear testable hypothesis stated?

-Is the study design appropriate to address the stated objectives?

-Is the population clearly described and appropriate for the hypothesis being tested?

-Is the sample size sufficient to ensure adequate power to address the hypothesis being tested?

-Were correct statistical analysis used to support conclusions?

-Are there concerns about ethical or regulatory requirements being met?

Reviewer #1: (No Response)

Reviewer #2: An retrospective epistudy of sera was collected from various areas in Spain from folks that had any clinical presentations included in a typical CCHF case definition. These sera were screened using various antigenic and nucleic methods (ELISA/PCR/Sanger etc.,) which revealed four retrospective cases of CCHF. The virus was interestingly typed to the European clade, suggesting that both circulating African clade and European clade viruses were present and infecting both ticks and humans within Spanish borders. 

Methods are appropriate for the study aims. This reviewer would recommend to including the type of reagents used in the in-house ELISA and IFA assay used for detection? What were the antigens or antibodies utilized as well as the methodologies?

**Results**

-Does the analysis presented match the analysis plan?

-Are the results clearly and completely presented?

-Are the figures (Tables, Images) of sufficient quality for clarity?

Reviewer #1: Results are clearly presented and decision workflow for positive and negative samples clearly laid out.

Reviewer #2: (No Response)

**Conclusions**

-Are the conclusions supported by the data presented?

-Are the limitations of analysis clearly described?

-Do the authors discuss how these data can be helpful to advance our understanding of the topic under study?

-Is public health relevance addressed?

Reviewer #1: Conclusions are supported.

Reviewer #2: (No Response)

**Editorial and Data Presentation Modifications?**

Reviewer #1: (No Response)

Reviewer #2: Accept, however, minor editing throughout would enhance clarity; this reviewer recommends having the manuscript submitted to an English/grammar proofreading service.

**Summary and General Comments**

Reviewer #1: Manuscript is well written with only minor editing for grammar and clarity needed. Authors interestingly identified a novel strain of CCHFV that until now was not known to be circulating in Spain. This conclusions is well supported by their phylogenetic analysis which strongly argues against this identified strain being a descendant of already known strains in Spain. Their data also suggests multiple introductions of CCHFV into Spain, an important finding that highlights the continued spread of CCHFV. 

Minor comments:

The positive acute case was associated with cattle husbandry and the virus clusters with Turkey, Kosovo and Russian strains of CCHFV. Is there any movement of cattle between these regions and does the patient have any history of travel to these regions? If not can authors speculate on how CCHFV from this region spread to Spain? Are there migratory birds that travel between these regions? Livestock trade? 

Patient population presented with an undifferentiated febrile illness and only one patient was positive for CCHFV. Do authors have any data on what the rest of the patients had? 

Line 288 is missing a reference. 

Authors state that their qRT-PCR was modified to include an internal control. What was this control?

Reviewer #2: This work represents very forward thinking clinical sciences. It is of great relevance to the geographic area in terms of understanding the past and likely future burden of clinical CCHF.

PLOS authors have the option to publish the peer review history of their article (what does this mean?). If published, this will include your full peer review and any attached files.

Reviewer #1: No

Reviewer #2: No
---

## [Editor Report · Decision Letter 1]

2 Feb 2021

Dear Dr. Belhassen-García,

We are pleased to inform you that your manuscript 'New circulation of genotype V of Crimean-Congo haemorrhagic fever virus in humans from Spain.' has been provisionally accepted for publication in PLOS Neglected Tropical Diseases.

Best regards,

Michael R Holbrook, PhD

Associate Editor

Dennis Bente

Deputy Editor

---

## [Editor Report · Acceptance letter]

17 Feb 2021

Dear Dr. Belhassen-García,

We are delighted to inform you that your manuscript, "New circulation of genotype V of Crimean-Congo haemorrhagic fever virus in humans from Spain.," has been formally accepted for publication in PLOS Neglected Tropical Diseases.

Best regards,

Shaden Kamhawi

co-Editor-in-Chief

Paul Brindley

co-Editor-in-Chief
